# A Flexible Approach for Assessing Heterogeneity of Causal Treatment Effects on Patient Survival Using Large Datasets with Clustered Observations

**DOI:** 10.3390/ijerph192214903

**Published:** 2022-11-12

**Authors:** Liangyuan Hu, Jiayi Ji, Hao Liu, Ronald Ennis

**Affiliations:** 1Department of Biostatistics and Epidemiology, Rutgers University, New Brunswick, NJ 07102, USA; 2Cancer Institute of New Jersey, Rutgers University, New Brunswick, NJ 07102, USA; 3Robert Wood Johnson Medical School, Rutgers University, New Brunswick, NJ 07102, USA

**Keywords:** causal inference, survival data analysis, machine learning, treatment effect heterogeneity, clustering

## Abstract

Personalized medicine requires an understanding of treatment effect heterogeneity. Evolving toward causal evidence for scenarios not studied in randomized trials necessitates a methodology using real-world evidence. Herein, we demonstrate a methodology that generates causal effects, assesses the heterogeneity of the effects and adjusts for the clustered nature of the data. This study uses a state-of-the-art machine learning survival model, riAFT-BART, to draw causal inferences about individual survival treatment effects, while accounting for the variability in institutional effects; further, it proposes a data-driven approach to agnostically (as opposed to a priori hypotheses) ascertain which subgroups exhibit an enhanced treatment effect from which intervention, relative to global evidence—average treatment effects measured at the population level. Comprehensive simulations show the advantages of the proposed method in terms of bias, efficiency and precision in estimating heterogeneous causal effects. The empirically validated method was then used to analyze the National Cancer Database.

## 1. Introduction

Knowing which subgroup of individuals could best benefit from a treatment is crucial to evidence-based medicine. It is critical to understand the treatment effect heterogeneity (TEH), which reflects the variability in patient response to treatment [1]. The population average treatment effect is less useful for personalized medicine because its estimate could potentially average out treatment benefits and harms.

In health research, patient survival is of most clinical relevance. There is a pressing need for robust causal inference methods to evaluate heterogeneous treatment effects on clustered patient survival. Existing methods for assessing heterogeneous treatment effects on patient survival are largely focused on subgroup analysis with a priori hypotheses about either subpopulations who might depart from the population average or interactions between treatment and pre-selected covariates. These methods are prone to multiple testing concerns and estimation bias [2]. More importantly, when patients are clustered, and when there are multiple treatments, special statistical considerations are needed to address the implications of the multilevel data structure for drawing causal inferences about multiple treatment comparisons.

## 2. Methods

We developed a random-intercept accelerated failure time model leveraging a probabilistic machine learning technique, Bayesian additive regression trees (BART) [3,4,5], for causal inferences about multiple treatments and clustered survival outcomes. This method, termed riAFT-BART [6], flexibly and accurately captures the relationships among the patient survival times, treatment and covariates via a sum of the tree models; it also accounts for the cluster-specific main effects using the random intercepts. Regularizing priors are placed on the parameters of riAFT-BART to ensure that the model is flexible in capturing nonlinearity and interactions but not overfitted [7,8]. An efficient and stable Markov chain Monte Carlo algorithm is developed for posterior inferences about the parameters. The formal statistical methodology was described in our earlier work [6] and in the Appendix A.

Putting riAFT-BART in the causal framework (Rubin causal model) [9,10], we can obtain the individual treatment effect by contrasting the counterfactual survival time to a treatment with the counterfactual survival time to another treatment (or control in a binary treatment setting) for each individual. These individual effects, bearing a causal interpretation, can then be used to identify TEH via a hypothesis-free and data-driven procedure, leveraging the Random Forest (RF) model. The “fit-the-fit” procedure [11,12] follows the following steps: (1) using the individual treatment effects as the responses, fit a sequence of RF models, where covariates are sequentially added in a stepwise manner to improve the model fit, as measured by $R^{2}$; (2) at each step, select the variable producing the largest $R^{2}$ improvement; (3) stop the fitting process when the percent improvement in $R^{2}$ is less than 1%. The final RF model will be interpreted using the “inTrees” technique [13]. The tree branch decision rules sending individuals to different end nodes define the combination rules of covariates that form different subpopulations having differentiable treatment effects. Subgroup treatment effects are estimated by averaging the individual effects in each end note. A detailed description of the method is provided in Hu (2022) [14]. 

## 3. Results

### 3.1. Data Examples

We first used simulated data to evaluate the proposed method. An expansive and representative simulation was conducted following the state-of-the-art guidance [6,10,15] on generating data adhering to the structure of multiple treatments with heterogeneous treatment effects on clustered survival outcomes. We based our simulation procedures on real data from the National Cancer Database (NCDB) [16]. We compared our proposed approach to three current methods popularly used in clinical research: (1) inverse probability of treatment weighting with the random-intercept Cox regression model (IPW-riCox) [9]; (2) doubly robust random-intercept additive hazards model (DR-riAH) [17]; and (3) the random-intercept generalized additive proportional hazards model (riGAPH) [18]. Complete details for the simulation design are presented in the Appendix A.

In both scenarios of proportional hazards (PH) and nonproportional hazards (nPH), our method, riAFT-BART, yielded the smallest biases (Appendix A) and root-mean-squared errors (Appendix A), and achieved the highest accuracy in estimating the the, indicated by the smallest PEHE values across all scenarios (Appendix A and Appendix A). The violation of the PH assumption had the least impact on the performance of our method but bore heavily on the random-intercept Cox regression model, which requires PH. Figure 1 shows the Kaplan–Meier survival curves for three simulated treatment groups, and the mean counterfactual survival curves estimated by the four methods under PH and nPH. The survival curves estimated by riAFT-BART are closest to the true survival curves for each treatment group under both PH and nPH, corroborating the accuracy of our proposed method in estimating the individual treatment effects (Table 1). 

### 3.2. Heterogeneous Treatment Effects for High-Risk Localized Prostate Cancer Patients

Next, we applied our method on 64,569 high-risk localized prostate cancer patients diagnosed between 2004 and 2015, drawn from the NCDB. We evaluated the TEH among three treatments: (i) radical prostatectomy (RP); (ii) external beam radiotherapy (EBRT) combined with androgen deprivation (AD) (EBRT + AD); and (iii) EBRT plus brachytherapy with or without AD (EBRT + brachy ± AD) [13]. Patients were naturally clustered within the institution. The pre-treatment risk factors included age, prostate-specific antigen (PSA), clinical T stage, Charlson–Deyo score, biopsy Gleason score, year of diagnosis, insurance status, median income level, education, race, and ethnicity. Appendix A summarizes the baseline characteristics of the patients. 

Appendix A demonstrates that, on average, the expected survival time for patients who underwent RP was 1.25 (1.15, 1.37) times as long as that of patients who underwent EBRT + brachy ± AD. However, among high-grade cancer patients with a Gleason score ≥ 9, there was no statistically significant treatment benefit associated with RP. When compared to EBRT + AD, RP led to a significantly longer survival time, and there was no directional TEH (Appendix A). Between EBRT + AD and EBRT + brachy ± AD, the population average treatment effect suggests a significant treatment benefit from EBRT + brachy ± AD; however, TEH analysis suggests that younger patients with lower PSA had a favorable treatment effect from EBRT + AD (Appendix A). Our method was able to identify the location (cluster-level) effects, which are displayed in Appendix A. Hospitals in New England had substantially better patient outcomes than hospitals in the South-Central area.

## 4. Discussion

We developed a machine learning-based method to evaluate the heterogenous causal effects on patient survival using real-world evidence data with clustered patient observations. This method provides a much-needed causal analysis tool for researchers to conduct in-depth analyses of multilevel data with a survival endpoint (overall and event-free survival). Our method can be used to gain insights into personalized treatment and institutional variation in treatment effects. Expansive and representative simulations provide strong empirical evidence that our method has better performance than existing methods in a wide range of data settings. Application to the NCDB data elucidates the importance of estimating heterogenous institutional and treatment effects. The developed methods provide an analysis apparatus for researchers working with large health datasets with clustered observations, and can aid in treatment effect discovery in subpopulations and inform the planning of future confirmatory trials.

## 5. Conclusions

For evidence-based medicine, it is difficult to apply evidence of population average effects to individual patients who might deviate from the population average. To reveal a potentially complex mixture of causal treatment effects (e.g., treatment benefit and treatment harm), robust analyses of TEH are needed. Large-scale clinical datasets collected from multiple institutions become common for generating real-world TEH evidence but introduce clustered observations. We developed a robust method to overcome these challenges and to provide accurate and efficient estimation of heterogenous causal treatment effects on patient survival. The TEH analysis of the NCDB prostate cancer data suggests that there may exist heterogeneous treatment effects between the EBRT-based treatments, which can inform individualized treatment strategies. The examination of variation due to clusters may stimulate further investigation into reasons why patient outcomes are different across clusters, which may lead to new insight into the quality of treatment delivery. Results from applying our methods to the NCDB prostate cancer data suggest a substantial variability in institutional effects, with hospitals in the New England area having much better patient outcomes than hospitals in the East Central region. Importantly, the results based on our methods are relevant to all stakeholders, including researchers, patients, clinicians and policymakers. For future research, an immediate extension of the proposed methods is to include the random slopes in the model so that the variability of the covariate effects across different clusters can be incorporated. Developing a sensitivity analysis to potentially have no unmeasured confounding factors could also be a worthwhile contribution.

## Figures and Tables

**Figure 1 ijerph-19-14903-f001:**
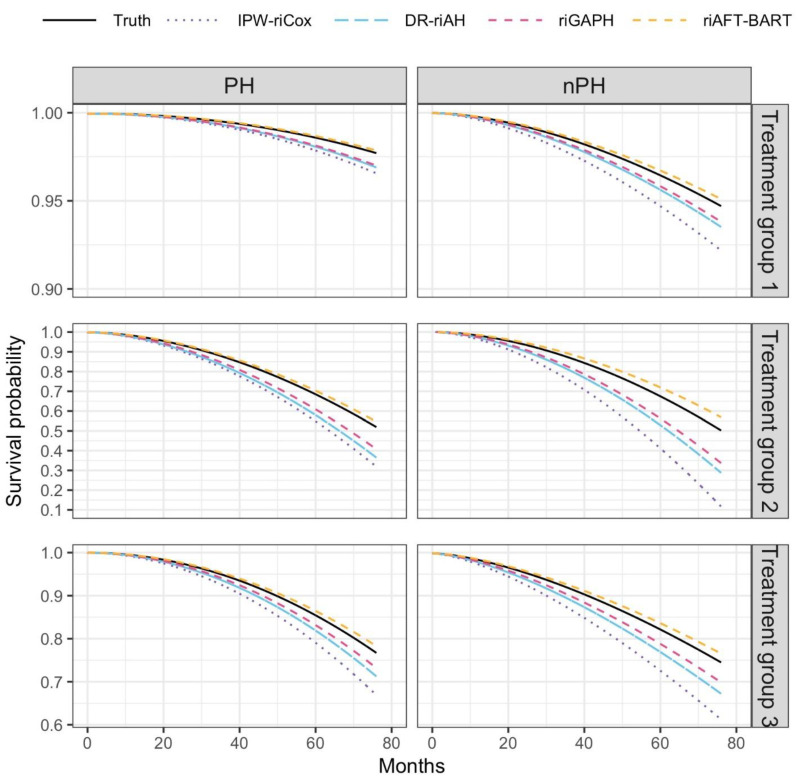
The Kaplan–Meier survival curves for three treatment groups in our simulation, and the mean counterfactual survival curves estimated by the four methods IPW-riCox, DR-riAH, riGAPH and riAFT-BART, under proportional hazards and nonproportional hazards.

**Table 1 ijerph-19-14903-t001:** Mean (and standard deviation) precision in the estimation of heterogeneous effects (PEHE) for each of the four methods based on 5-year survival probability. The proposed method riAFT-BART delivered the highest accuracy in estimating the treatment effect heterogeneity, indicated by the smallest PEHE values across all simulation settings.

	Proportional Hazards			Nonproportional Hazards		
Method	Trt 1 vs. 2	Trt 1 vs. 3	Trt 2 vs. 3	Trt 1 vs. 2	Trt 1 vs. 3	Trt 2 vs. 3
IPW-riCox	0.093 (0.032)	0.070 (0.030)	0.075 (0.031)	0.112 (0.033)	0.090 (0.030)	0.094 (0.031)
DR-riAH	0.043 (0.024)	0.022 (0.023)	0.027 (0.022)	0.051 (0.022)	0.034 (0.021)	0.039 (0.022)
riGAPH	0.041 (0.022)	0.020 (0.021)	0.025 (0.022)	0.049 (0.023)	0.031 (0.023)	0.038 (0.023)
riAFT-BART	0.013 (0.011)	0.006 (0.009)	0.008 (0.009)	0.018 (0.014)	0.011 (0.011)	0.013 (0.012)

## Data Availability

The proposed method can be implemented through the R package riAFT-BART, freely available on CRAN. The NCDB data used in the case study is publicly available upon approval of the NCDB Participant User File application.

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
