# Peer review of "A Flexible Approach for Assessing Heterogeneity of Causal Treatment Effects on Patient Survival Using Large Datasets with Clustered Observations"

_ijerph, 2022, doi:10.3390/ijerph192214903_

Round 1

Reviewer 1 Report

1.     This is a meaningful study as this brief report contributes to furthering the analysis of assessing the heterogeneity of causal treatment effects on patient survival.

2.     The research is valuable due to the method of using large datasets with clustered observations.

3.     The title and abstract are appropriate for the content of the brief report.

4.     The brief report is well organized.

5.     The discussion could be strengthened by using the results of the report as a reference for the field.

6.     The conclusion could be strengthened by adding some specific statements about the findings of this study.

Reviewer 2 Report

Dear Authors,

You have a statement "Expansive and representative simulations provide strong empirical evidence that our method has better performance than existing methods in a wide range of data settings." in the "Discussion" section. Could you please clarify (with the references) which of the existing methods have you compared your method with?

Please also add a couple lines of "Future Directions" to the end of "Conclusion" section.
